# A Small Highly Sensitive Glucose Sensor Based on a Glucose Oxidase-Modified U-Shaped Microfiber

**DOI:** 10.3390/s24020684

**Published:** 2024-01-21

**Authors:** Tingkuo Chen, Haiming Jiang, Kang Xie, Hongyan Xia

**Affiliations:** 1State Key Laboratory of Precision Electronic Manufacturing Technology and Equipment, School of Electromechanical Engineering, Guangdong University of Technology, Guangzhou 510006, China; chentingkuo@163.com; 2Department of Applied Sciences, Northumbria University, Newcastle upon Tyne NE1 8ST, UK

**Keywords:** glucose sensor, glucose oxidase, U-shaped microfiber, surface modification, evanescent wave

## Abstract

Diabetes patients need to monitor blood glucose all year round. In this article, a novel scheme is proposed for blood glucose detection. The proposed sensor is based on a U-shaped microfiber prepared using hydrogen-oxygen flame-heating technology, and then 3-aminopropyltriethoxysilane (APTES) and glucose oxidase (GOD) are successively coated on the surface of the U-shaped microfiber via a coating technique. The glucose reacts with the GOD of the sensor surface to produce gluconic acid, which changes the effective refractive index and then shifts the interference wavelength. The structure and morphology of the sensor were characterized via scanning electron microscope (SEM) and confocal laser microscopy (CLM). The experimental results show that the sensitivity of the sensor is as high as 5.73 nm/(mg/mL). Compared with the glucose sensor composed of the same material, the sensitivity of the sensor increased by 329%. The proposed sensor has a broad application prospect in blood glucose detection of diabetic patients due to the advantages of miniaturization, high sensitivity, and good stability.

## 1. Introduction

Diabetes has become one of the most deadly chronic diseases in modern society [1,2,3]. Its main symptom is pancreatic dysfunction, resulting in inadequate or inappropriate insulin secretion, and eventually leading to elevated blood glucose in diabetic patients [4,5]. If the blood glucose level is maintained at a high level for a long time, advanced diabetes causes a variety of complications and even life-threatening problems, such as heart disease, kidney disease, eye disease, nerve disease, and vascular necrosis [6,7]. At present, the main commercial method of blood glucose quantitative detection is an electrochemical method. This method has the characteristics of complex operation and long analysis time, which makes it difficult for patients to complete this operation at home [8,9]. An optical fiber sensor, which has the advantages of anti-electromagnetic interference, real-time monitoring, fast response, simple operation, and convenient production, can act as an attractive alternative to solve the above problem in blood glucose detection [10,11,12,13]. Particularly, since the outbreak of COVID-19 in December 2019, many people have been infected around the world [14,15]. This poses a serious obstacle to blood glucose monitoring and treatment of diabetes during the pandemic, as home isolation is required in medium-high-risk areas. To address this problem, the requirement of blood glucose testing in real time has been proposed to avoid the spread of COVID-19 [16].

Compared with the linear fiber sensor, the U-shaped fiber sensor shows higher sensitivity to the surrounding refractive index change [17,18]. This is because U-shaped fiber has larger evanescent power and penetration depth, forming a stronger interaction between evanescent wave and functional material [19,20]. In addition, the U-shaped sensor is a traditional probe structure, which is more convenient to carry or detect [21,22,23]. U-shaped fiber is classified into U-shaped microfiber (≤10 μm) and U-shaped fiber (>10 μm). U-shaped microfiber inherits most advantages of U-shaped optical fiber and microfiber, including high sensitivity, strong evanescent wave, miniaturization, flexibility, and so on [24,25,26]. In the past five years, optical fiber sensors based on U-shaped microfiber have been greatly developed. In 2018, Zhao et al. constructed a humidity sensor by coating asymmetric U-shaped microfiber with polyvinyl alcohol (PVA), which has a sensitivity of 0.186 nm/%RH in the humidity range of 30%RH–95%RH [27]. The following year, the group built a humidity sensor with an asymmetric U-shaped microfiber coupler that featured an ultra-wide detection range (18%RH–95%RH) [28]. In 2021, Liu et al. designed a humidity sensor using graphene oxide (GO)-coated U-shaped microfiber with a sensitivity of up to 0.361 nm/%RH [29]. As can be seen from the previous works, a U-shaped microfiber sensor not only has high sensitivity but also has a wide detection range.

In order to improve the sensitivity of the sensor, it is most common that the surface of the optical fiber sensor is coated with functional materials [30,31]. Among them, glucose oxidase (GOD) is regarded as the preferred functional material in the field of glucose sensing [32] because it not only increases the sensitivity but also shows the selectivity of glucose detection [33,34]. The sensing principle is as follows: GOD can combine with glucose to produce hydrogen peroxide and gluconic acid, which will cause the refractive index of the coating layer on the surface to change, so as to affect the optical signal in the fiber [35,36]. In 2019, Ding et al. coated a glucose oxidase/carbon quantum dots/cellulose acetate (GOD/CQDs/CA) hybrid material on the fiber end to construct a glucose fluorescence sensor, with a minimum detection limit of 25.79 nM in the range of 10–100 nmol/L [37]. The next year, the group came up with a scheme using GO and GOD to immobilize long-period gratings; the sensitivity of the glucose sensor (concentration: 0–1.2 mg/mL) is 0.77 nm/(mg/mL) [38]. In 2022, Chen’s group constructed a surface plasmon resonance sensor using glucose oxidase/polystyrene (GOD/PS) as a coated functional material. The glucose sensor has a sensitivity of 0.31 nm/(mg/mL) in the 0–4 mg/mL concentration [39]. However, these sensors still have problems such as low sensitivity and large volume.

In this paper, we propose a small and highly sensitive glucose sensor based on a U-shaped microfiber. The sensing performance of the non-functionalized sensor and the sensor was tested to ensure that the sensitivity gain from the coating was real and effective. The experimental results show that the average sensitivities of the non-functionalized sensor and the sensor were 2.26 nm/(mg/mL) and 5.73 nm/(mg/mL), respectively. We characterized the stability, temperature cross-sensitivity, selectivity, and dynamic response of the sensor. Compared with the glucose sensor composed of the same material, the sensitivity, stability, and portability of the sensor are greatly improved. Among them, the sensitivity is improved by 329%.

## 2. Materials and Methods

### 2.1. Materials

Glucose oxidase (GOD, 99%) was procured from Aladdin (Shanghai, China). 3-Aminopropyltriethoxysilane (APTES, 98%) was provided from Beyotime (Shanghai, China). Glucose (99%), fructose (99%), Nacl (99.5%), Kcl (99.5%), cholesterol (98%), and anhydrous ethanol (99.5) were all purchased from Macklin (Shanghai, China). Saccharose (99%), urea (99%), $H_{2}{SO}_{4}$ (99.9%), and $H_{2}O_{2}$ (30%) were offered by Guangzhou Chemical Reagent Factory (Guangzhou, China). PBS (99%) was offered by Xiamen Haibiao Technology Co., Ltd. (Xiamen, China). SMF (G.652.D) was received from YOFC (Wuhan, China). Fetal Bovine Serum (100%) and Dulbecco’s Modified Eagle Medium (100%) were provided by Gibco.

### 2.2. U-Shaped Microfiber Fabrication and Surface Functionalization

The microfiber used in this study was prepared using an optical fiber pulling mechanism, among which the important components of the instrument include a hydrogen and oxygen lamp heating device and a displacement platform stretching device. A standard single-mode fiber was heated and elongated to 1.2 cm at a uniform rate to obtain a microfiber with a diameter of approximately 7 µm. The microfiber was then immobilized in a capillary glass tube by using a UV photoresist, and a U-shaped microfiber with a radius of about 2 mm was obtained. As shown in Figure 1 below, the specific process of the functionalization of the U-shaped microfiber was conducted as follows. First, the U-shaped microfiber was immersed in piranha solution (7:3 = ${SO}_{4}$:$H_{2}O_{2}$) for 1 h to activate the surface hydroxyl group (-OH). Then, the U-shaped microfiber was filled with 10% APTES solution and kept for 20 min to deposit APTES on its surface to form amino groups (-NH3) [40]. Finally, the siliconized U-shaped microfiber was incubated for 2 h in GOD solution (4 mg/mL), and thus GOD was fixed on the surface of the U-shaped sensing region [41].

### 2.3. Characterizations of the Sensor

The morphology of the sensor was observed via focusing ion beam field emission scanning electron microscopy (Tescan, LYRA 3 XMU, Prague, Czech) and a laser confocal microscope (Carl Zeiss, LSM800, Berlin, Germany). The surface of the sensor was sputtered with a layer of gold using an ion sputtering machine (Vacuum, MSP-1S, Tokyo, Japan) before the sample was observed using an electron microscope. At the same time, the sensing area of the sensor was observed using a laser confocal microscope, whose excitation light was 488 nm.

### 2.4. Experimental System

The optical system for glucose detection is shown in Figure 2. A supercontinuous light source (YSL, SC-5, Wuhan, China) provided a 1500–1600 nm light source signal for the optical system, and a spectral analyzer (Yokogawa, AQ6370D, Tokyo, Japan) with a minimum resolution of 20 pm was used to receive the optical signal from the sensor. Compared with the blood glucose range of healthy people (0.5–1.2 mg/mL), this study was mainly aimed at measuring blood glucose in diabetic people. Therefore, the concentrations of glucose samples to be measured were set to 0, 1.0, 2.0, and 3.0 mg/mL. To investigate the temperature cross-sensitivity of the sensor, a 3.0 mg/mL glucose solution was heated to 34, 35, 36, 37, 38, 39, and 40 °C using a heating device (JINFENG, JF-956s, Dongguan, China). When detecting the dynamic response of the sensor, the bench light source of 1550 nm was used as the input light, the photodetector (EOT, ET-3500F, Traverse, MI, USA) was used as the detection of the change in the optical signal power, and the oscilloscope (RIGOL, MSO8000, Suzhou, China) was used as the reading and analysis equipment of the electrical signal.

### 2.5. Theoretical Analysis

As light travels through the sensor, light leaks out of the microfiber in the U-shaped region, resulting in evanescent waves on the surface of the microfiber. The evanescent wave can be used to sense the surrounding medium, and the detection sensitivity can be improved by coating the microfiber surface with a functional material. When the sensor coated with APTES and GOD is placed in the glucose solution, the following reactions occur on the surface of the microfiber in the U-shaped region of the sensor [42]:(1)$$Glucose+H_{2}O+O_{2}\overset{GOD}{\to }Glucose\;Acid+H_{2}O_{2}$$

In the reaction (Equation (1)), glucose, water, and oxygen are catalyzed by GOD to produce gluconic acid and hydrogen peroxide. The gluconic acid changes the ambient refractive index RI of the sensor, which affects the optical signal in the microfiber [43]. Compared with other mode fields, the coupling strength between ${HE}_{11}$ and ${HE}_{12}$ is the most obvious in the microfiber. The formula for optical wavelength and environmental refractive index of the sensor is [44]:(2)$$\frac{d\lambda }{dn_{ex}}=\frac{\lambda }{\Gamma }(\frac{1}{\Delta n}\frac{d\Delta n}{dn_{ex}})$$
where λ is the optical wavelength, $n_{ex}$ is the environmental refractive index, and Δn is the effective refractive index difference. $\Gamma =1-\frac{\lambda }{\Delta n}\frac{d\Delta n}{d\lambda }$ is the dispersion factor, which is usually negative because of the waveguide properties of the microfiber. Based on Equation (2), when the environmental refractive index increases, the index increment of the ${HE}_{12}$ mode is larger than that of the ${HE}_{11}$ mode. Thus, $d\Delta n/dn_{ex}$ obtain a negative value [45]. The results show that the interference wavelength of light redshifts as the ambient refractive index increases.

In the course of this study, the elevation of glucose solution concentration from 0 to 3 mg/mL induces an increase in the refractive index of the environment surrounding the sensor. In the microfiber, the coupling strength between different mode fields results in a redshift in the interference wavelength [46,47]. This phenomenon constitutes the fundamental principle underlying the glucose detection mechanism of the APTES/GOD-functionalized microfiber sensor.

## 3. Results

### 3.1. Morphology Analysis of the Sensor

Following the successful functionalization of the microfiber surface, detailed SEM characterization was conducted to elucidate both the structure of the microfiber and the coating material APTES. Figure 3A illustrates the structure of the sensor: it is a U-shaped microfiber with a radius of 400 µm. For a comprehensive understanding of the sensing region, Figure 3B,C present SEM images of the non-functionalized and APTES-functionalized microfibers, respectively. The discernible diameters of the non-functionalized and functionalized microfibers are approximately 7.3 µm and 7.5 µm, respectively. These observations signify the successful application of a uniform 0.2 µm thick coating on the microfiber’s surface. This critical step enhances the sensor’s sensitivity in glucose sensing, underscoring its effectiveness. In the next step, to ascertain the secure immobilization of glucose oxidase (GOD) on the APTES-functionalized material, fluorescence detection was employed. The sample underwent irradiation with a 488 nm laser using a laser confocal microscope. Figure 3 depicts microscopic images of APTES-functionalized microfibers in Figure 3D, along with fluorescence images of APTES-functionalized microfibers in Figure 3E and APTES/GOD-functionalized microfibers in Figure 3F. In comparison with the fluorescence images of the APTES-functionalized microfiber, the fluorescence signal of the APTES/GOD-functionalized microfiber shows a marked enhancement, providing clear confirmation of the successful immobilization of GOD molecules onto the APTES-functionalized surface. The fundamental principle of the functionalization process is as follows: initially, APTES is deposited onto the microfiber’s surface, concurrently immobilizing free amino groups (−NH3) on the APTES surface. Subsequently, during incubation with the GOD solution, fixation is accomplished through the binding of GOD’s carboxyl groups (−COOH) with the amino groups on the APTES surface. In summary, the integration of APTES serves as a critical step in enhancing the structural and functional attributes of the microfiber surface, ultimately contributing to heightened sensitivity and the successful immobilization of GOD, which is pivotal for the sensor’s efficacy in glucose detection.

### 3.2. Sensing Performance of the Sensor

To assess the sensing performance of the glucose detection sensor, various concentrations of glucose solutions were tested. Specifically, the concentrations tested were 0, 1, 2, and 3 mg/mL, with the interference wavelength recorded and analyzed using a spectrometer. As shown in Figure 4A, with the gradual increase in glucose solution concentration, the interference wavelength of the sensor was redshifted. It shifted from 1555.0 nm to 1572.2 nm for a total shift of 17.2 nm. Consistent with the results of theoretical analysis (Equation (2)), the interference wavelength redshifts with the increase in glucose concentration. As shown in Figure 4B, the linear fitting sensitivity of the sensor was 5.73 nm/(mg/mL). Moreover, the $R^{2}$ of the fitted line is 0.994; therefore, the sensor has excellent linearity. At the same time, in order to prove that APTES and GOD play an important role in the sensing performance of the U-shaped microfiber, we used the non-functionalized sensor to detect the same glucose solution. As shown in Figure 4C, the interference wavelength of the non-functionalized sensor was also redshifted. It shifted from 1569.8 nm to 1576.6 nm for a total shift of 6.8 nm. As shown in Figure 4D, the linear fitting sensitivity of the non-functionalized sensor is 2.26 nm/(mg/mL). According to the comparison of the non-functional sensor, the sensitivity of the sensor is improved by 2.53 times. Therefore, it can be concluded that APTES and GOD materials provide a significant improvement in the sensing sensitivity of the non-functionalized sensor.

To assess the practicality of this sensor, detections were also conducted using a complete growth medium. The complete growth medium, composed of 90% Dulbecco’s Modified Eagle Medium and 10% Fetal Bovine Serum, is commonly employed for cell cultivation and encompasses most of the nutritional components found in the blood. Therefore, substituting blood with a complete growth medium is justifiable. As illustrated in Figure 5, with increasing glucose concentrations, a redshift in the interference wavelength was observed. A total shift of 12.7 nm was recorded, yielding a sensitivity of 3.13 nm/(mg/mL). Notably, this sensitivity is lower than the ideal sensitivity of 5.73 nm/(mg/mL), which could be attributed to the influence of various complex components present in the complete growth medium.

Table 1 shows the glucose sensors coated with GOD using optical devices with different structures, including microfiber, fiber surface plasmon resonance (SPR), tilted fiber grating (TFG), and side-hole fiber SPR. Compared with sample 2 and sample 3, sample 1 has greater sensitivity, indicating that microfiber can improve the sensitivity of glucose sensors. Compared with other samples, the sensitivity of sample 4 increased by more than one order of magnitude due to the presence of side-hole fiber SPR. However, its narrow range of detection (0–0.3 mg/mL) makes it difficult to afford a full range and effective detection of blood glucose. In view of the above situation, a sensing scheme of the U-shaped microfiber coated with GOD is proposed in this study. The sensitivity of the sensor is 5.73 nm/(mg/mL) in the concentration range of 0–0.3 mg/mL. Compared with sample 1 composed of the same material, the sensitivity improved by 329% under the same concentration range. The Limit of Detection (LOD) for the sensor in this study is 0.17 mg/mL. In comparison with sample 4, this sensor does not exhibit a superior LOD.

### 3.3. Stability and Temperature Cross-Sensitivity Analysis of the Sensor

Having good stability is a key indicator of a sensor, and we developed a correlation characterization (Figure 6). A 3.0 mg/mL glucose solution was detected using the sensor and recorded at one-minute intervals (10 min total). In the range of 1550–1580 nm, the maximum fluctuation of the interference wavelength is 1.2 nm, and the interference displacement only accounts for 7% of the total displacement. The results show that the sensor has good stability in the studied time span.

In addition to stability, temperature cross-sensitivity is also an important parameter of the sensor. The interference wavelength of the sensor shifted due to the change in the refractive index of the sensor surface, which requires the external ambient temperature to remain stable. Due to the photothermal effect of the optical fiber itself, the change in ambient temperature can make the refractive index change, which eventually affects the detection of the sensor. As shown in Figure 7A, as the temperature rises from 34 °C to 40 °C, the maximum blue shift occurs in the 1560-1580 nm band. As shown in Figure 7B, the interference wavelength moves from 1568.8 nm to 1567.4 nm, with a total translation of 1.4 nm. Compared with the total displacement caused by glucose sensing, the displacement caused by temperature accounted for only 8% [8]. Thus, the sensor can work normally in the range of 34 to 40 °C, and the temperature cross-sensitivity of the sensor is negligible.

### 3.4. Specificity and Dynamic Response Analysis of the Sensor

The proposed sensor is mainly aimed at the detection of diabetic blood samples; there are many substances in blood samples that interfere with sensing. To evaluate the specific detection ability of the sensor in complex partitioning, we selected multiple solutions with the same concentration as the interference components (including fructose, urea, sucrose, cholesterol, sodium chloride, potassium chloride, and calcium chloride). As shown in Figure 8, the average wavelength shift reached 15.01 nm at a glucose concentration of 3 mg/mL. Compared with other interfering components, the sensor has a high specific detection ability for glucose. In addition, fructose and cholesterol also caused significant wavelength shifts compared with other components (except glucose). In the case of fructose, it could be that fructose has a similar molecular structure to glucose. For cholesterol, cholesterol solutions belonging to lipids have a great change in refractive index.

In addition, the response time of the sensor was accurately evaluated, and we built an optical response test system (Figure 9A). For rapid response of the sensor, the concentration of the detected glucose solution was set to 3.0 mg/mL. As shown in Figure 9B, the response time should be calculated from 10% to 90% of the maximum response value of the signal; the response time of the sensor is about 0.264 s, which is faster than a similar type of glucose sensor [38].

## 4. Conclusions

In this work, a small, highly sensitive glucose sensor with a stable structure, simple fabrication, and low cost is proposed. The experimental results show that the sensitivity of the sensor for glucose (0–3.0 mg/mL) detection is 5.73 nm/(mg/mL). In addition, the errors caused by the stability and temperature cross-sensitivity of the sensor are negligible. Moreover, the sensor also has an ultra-short response time (0.264 s) and high specificity detection ability. Therefore, the detection scheme of the U-shaped microfiber modified by APTES and GOD provides a great reference value for medical diagnosis.

## Figures and Tables

**Figure 1 sensors-24-00684-f001:**
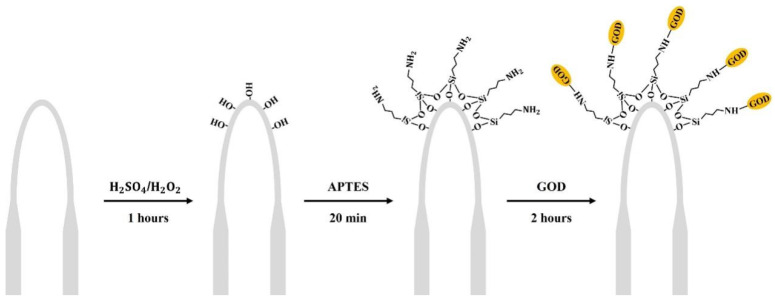
Schematic diagram of the functionalization of the glucose sensor.

**Figure 2 sensors-24-00684-f002:**
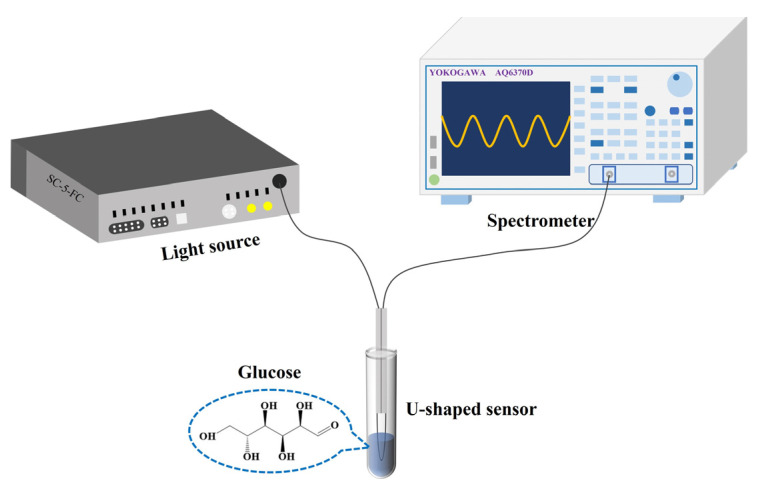
Schematic illustration of experiment setup for measuring glucose with the sensor.

**Figure 3 sensors-24-00684-f003:**
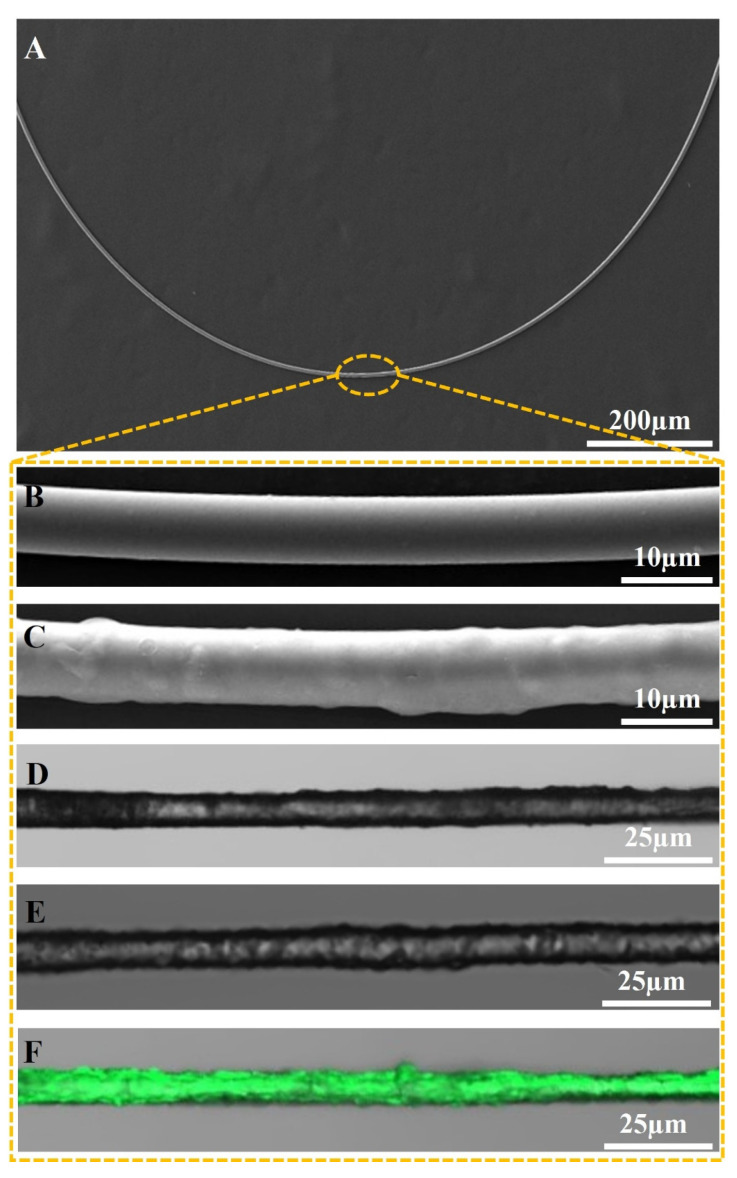
SEM images of (**A**,**B**) bare microfiber and (**C**) APTES-functionalized microfiber; microscope images of (**D**) APTES-functionalized microfiber and (**E**) fluorescence image of APTES-functionalized microfiber; and (**F**) fluorescence image of APTES/GOD-functionalized microfiber.

**Figure 4 sensors-24-00684-f004:**
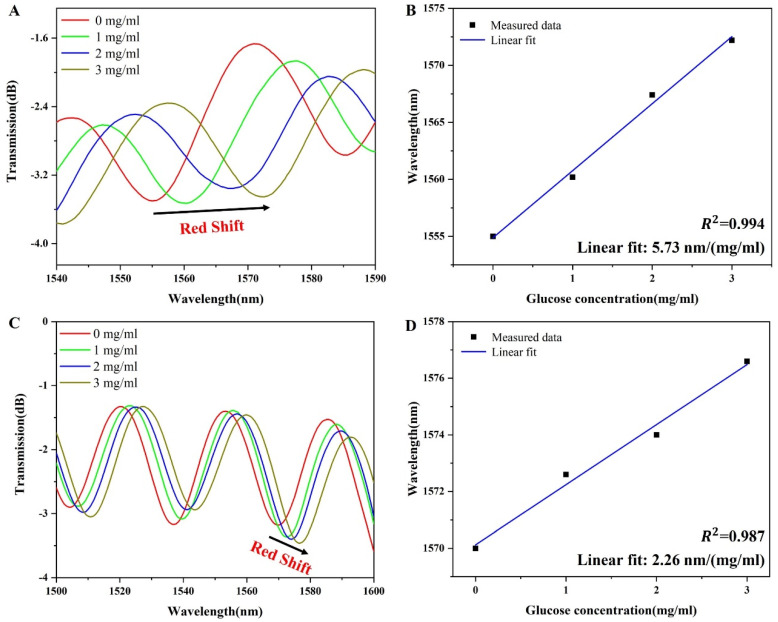
(**A**) Spectra of the sensor with the glucose concentration, (**B**) the linear fitting graph of the glucose concentration and the interference wavelength of the sensor; (**C**) spectra of the non-functionalized sensor with the glucose concentration, and (**D**) the linear fitting graph of the glucose concentration and the interference wavelength of the non-functionalized sensor.

**Figure 5 sensors-24-00684-f005:**
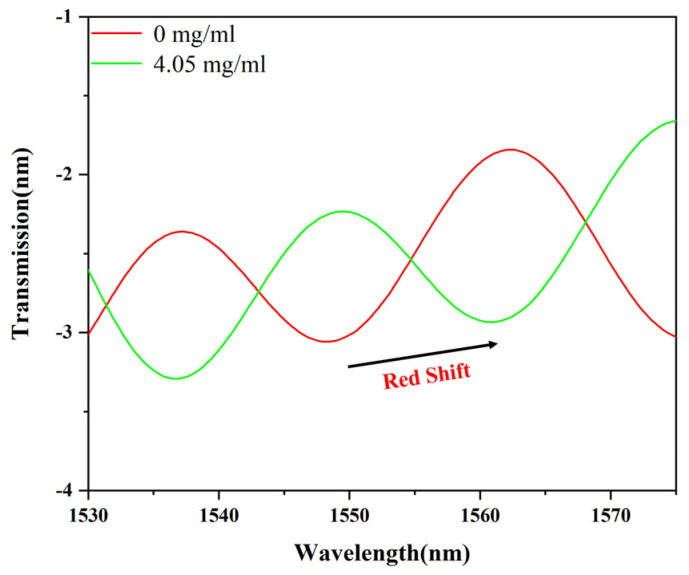
Spectra of complete growth medium (4.05 mg/mL) and glucose solution (0 mg/mL).

**Figure 6 sensors-24-00684-f006:**
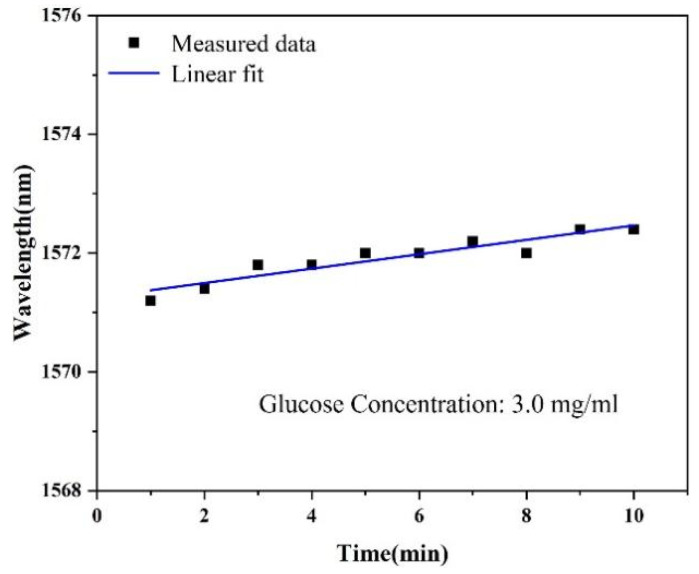
The relationship between time and wavelength shift.

**Figure 7 sensors-24-00684-f007:**
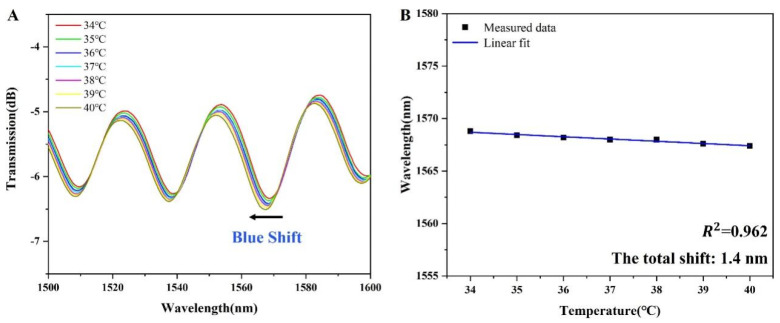
(**A**) Temperature experiment measurement spectrum diagram; (**B**) temperature sensitivity fitting diagram.

**Figure 8 sensors-24-00684-f008:**
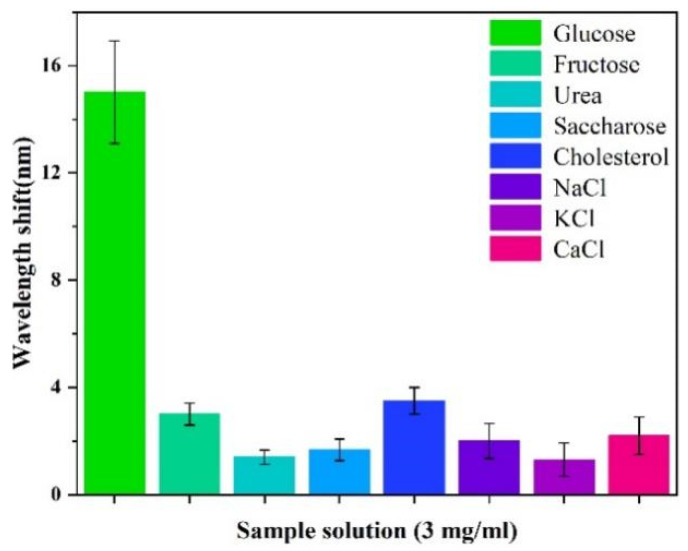
Selectivity with the interfering substances concentrations of 3.0 mg/mL.

**Figure 9 sensors-24-00684-f009:**
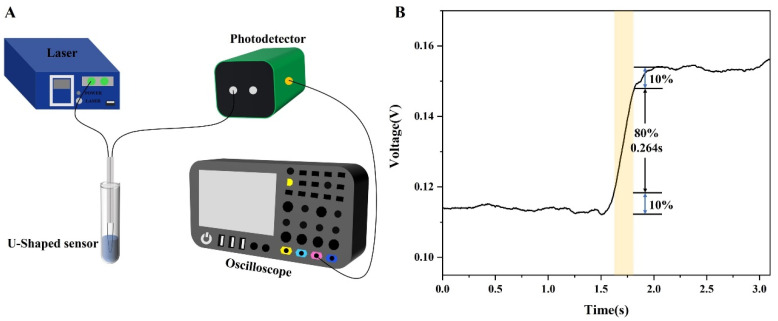
(**A**) Schematic diagram of an experimental device for detecting response time; (**B**) response time of the sensor.

**Table 1 sensors-24-00684-t001:** Comparison of sensing performance between different optical fiber glucose sensors.

Sample	Sensing Element	Glucose Sensitivity (nm/(mg/mL))	Measurement Range (mg/mL)	LOD	Ref.
1	Microfiber with GOD	1.74	0–3	/	[48]
2	Fiber SPR with PS/GOD	0.31	0–4	/	[39]
3	TFG with GO/GOD	1.34	0–1.5	/	[49]
4	Side-hole fiber SPR with GOD	34.20	0–0.3	16.24 μM	[50]
5	U-shaped microfiber with GOD	5.73	0–3	0.17 mg/mL	This work

## Data Availability

The dataset generated and analyzed during this study is available from the corresponding author upon reasonable request, but restrictions apply to the commercially confident details.

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
