# Peer review of "A Small Highly Sensitive Glucose Sensor Based on a Glucose Oxidase-Modified U-Shaped Microfiber"

_sensors, 2024, doi:10.3390/s24020684_

Round 1

Reviewer 1 Report

Comments and Suggestions for Authors

The authors should show the intrinsic fluorescence intensity of GOD before and after functionalization and compare them.

Commonly APTES is used as linker for functionalization of different agents on the surface. The role of APTES in observed results should be discussed and interpret. 

LOD of proposed method should be indicated and compare with other approaches in table 1. 

Comments on the Quality of English Language

Authors should check the manuscript to make sure the high quality of english language of manuscript. 

Reviewer 2 Report

Comments and Suggestions for Authors

The manuscript describes a small highly sensitive glucose sensor based on glucose oxidase modified U-shaped microfiber. The newly developed sensor was used for the determination of glucose. In my opinion, this paper may be accepted in the Journal of Sensor after major modification on the following points:

- -In the section " Materials ", the purity of materials used should be mentioned in this section.

-The author should give a reference for the procedure of the Surface functionalization of the sensor.

-The relation between wavelength and concentration is not clear in this study, so the author should give more details with related references in the text.

-To assess the applicability of the sensor, the author should determine the concentration of glucose in blood samples using the new sensor.

Reviewer 3 Report

Comments and Suggestions for Authors

This is an excellent work. It can be accepted directly.

Round 2

Reviewer 1 Report

Comments and Suggestions for Authors

I recommend publication of revised version of manuscript. 

Reviewer 2 Report

Comments and Suggestions for Authors

Accept in present form.